



**A long-term high-resolution air quality reanalysis with public facing air quality dashboard**
**over the Contiguous United States (CONUS)**
Rajesh Kumar[1], Piyush Bhardwaj[1,*], Cenlin He[1], Jennifer Boehnert[1], Forrest Lacey[1], Stefano
Alessandrini[1], Kevin Sampson[1], Matthew Casali[1], Scott Swerdlin[1], Olga Wilhelmi[1], Gabriele G.
Pfister[1], Benjamin Gaubert[1], and Helen Worden[1]
[1]NSF National Center for Atmospheric Research, Boulder, CO, USA
*Now at Center for Study of Science, Technology and Policy (CSTEP), Bengaluru, India
Corresponding author: Rajesh Kumar (rkumar@ucar.edu)
Keywords: Chemical data assimilation, WRF-CMAQ, GSI, Air Quality Dashboard



**Abstract**
We present a 14-year 12-km hourly air quality dataset created by assimilating satellite observations
of aerosol optical depth (AOD) and carbon monoxide (CO) in an air quality model to fill gaps in
the contiguous United States (CONUS) air quality monitoring network and help air quality
managers understand long-term changes in county level air quality. Specifically, we assimilate the
Moderate Resolution Imaging Spectroradiometer (MODIS) AOD and the Measurement of
Pollution in the Troposphere (MOPITT) CO observations in the Community Multiscale Air
Quality Model (CMAQ) every day from 01 Jan 2005 to 31 Dec 2018 to produce this dataset. The
Weather Research and Forecasting (WRF) model simulated meteorological fields are used to drive
CMAQ offline and to generate meteorology dependent anthropogenic emissions. Both the weather
and air quality (surface fine particulate matter ($PM_{2.5}$) and ozone) simulations are subjected to a
comprehensive evaluation against multi-platform observations to establish the credibility of our
dataset and characterize its uncertainties. We show that our dataset captures regional hourly,
seasonal, and interannual variability in meteorology very well across the CONUS. The correlation
coefficient between the observed and simulated surface ozone and $PM_{2.5}$ concentrations for
different Environmental Protection Agency (EPA) defined regions across CONUS are 0.77-0.91
and 0.49-0.79, respectively. The mean bias and root mean squared error for modeled ozone are
3.7-6.8 ppbv and 7-9 ppbv, respectively, while the corresponding values for $PM_{2.5}$ are -0.9-5.6
$\mu g/m^3$ and 3.0-8.3 $\mu g/m^3$, respectively. We estimate that annual CONUS averaged maximum daily
8-hour average (MDA8) ozone and $PM_{2.5}$ trends are -0.30 ppb/year and -0.24 $\mu g/m^3$/year,
respectively. Wintertime MDA8 ozone shows an increasing but statistically insignificant trend at
several sites. We also found a decreasing trend in the 95[th] percentile of MDA8 ozone but an
increasing trend in the 5[th] percentile. Most of the sites in the Pacific Northwest show an increasing



but statistically insignificant trend during summer. An ArcGIS air quality dashboard has been
developed to enable easy visualization and interpretation of county level air quality measures and
trends by stakeholders, and a Python-based Streamlit application has been developed to allow the
download of the air quality data in simplified text and graphic formats.



## 1.    Introduction

Air quality is one of the most important global environmental concerns as almost the entire global
population (99%) is estimated to breathe air that exceeds the World Health Organization (WHO)
defined Air Quality Guidelines (WHO, 2023). Exposure to ambient air pollution causes about 4.2
million premature mortalities every year (WHO, 2020). Air quality has improved substantially
over the past two decades in the US as the Environmental Protection Agency (EPA) observations
show that maximum daily 8h average (MDA8) surface ozone levels have decreased by 29% over
1980-2021, and annual average concentrations of particulate matter with an aerodynamic diameter
smaller than 2.5 μm ($PM_{2.5}$) have decreased by 37% over 2000-2021 (https://www.epa.gov/air-
trends/air-quality-national-summary). However, air pollution continues to violate the National
Ambient Air Quality Standards (NAAQS) in many parts of the US, such as the Colorado Front
Range, California, northeast US, and nearly all the national parks. A recent study reported that
97% of US national parks suffer from significant or unsatisfactory levels of harm from air pollution
(Orozco et al., 2024). Poor air quality is reported to cause about 160,000 premature deaths in the
US, with a total economic loss of about $175 billion (Im et al., 2018). Exposure to air pollution
levels even below the EPA NAAQS can adversely affect human health (Di et al., 2017). To
mitigate the risks of air pollution and how air quality is responding to emission control policies, it
is, therefore, imperative to quantify past changes in air quality.

64         Numerous studies have revealed several key features of long-term changes in surface ozone

and $PM_{2.5}$ over the US using long-term observations from the EPA monitoring networks. First,
both the urban and rural sites in the eastern US show negative ozone trends during the summer
season (Butler et al., 2011; Cooper et al., 2012), but lower ozone levels at some sites have an
increasing trend during winter and early spring (Bloomer et al., 2010; Cooper et al., 2012; Simon





et al., 2015). Second, surface and free tropospheric ozone show positive trends in all seasons at
rural and remote sites in the western US (Jaffe and Ray, 2007; Cooper et al., 2012). Third,
increasing ozone is observed in the inflow to the US west coast (Jaffe et al., 2003), over the North
Pacific (Parrish et al., 2004), and west coast marine boundary layer (Parrish et al., 2009). The
Tropospheric Ozone Assessment Report (TOAR) showed that summertime surface ozone
continues to decrease over the US, but the trend is less certain at the urban sites (Chang et al.,
2017; Fleming et al., 2018). Similar regional and seasonal differences in the long-term trends are
also seen in $PM_{2.5}$ and its components. For example, carbonaceous aerosols (organic and black
carbon) show a widespread decrease over 1990-2010 across the US in winter and spring and show
positive but insignificant trends over the western US (Hand et al., 2013). $PM_{2.5}$ levels continue to
decrease over the majority of the US except in the wildfire-prone areas (McClure and Jaffe, 2018).
In addition to the observation-based trend analysis, chemical transport model (CTM)
simulations have been employed to interpret the observed trends. For example, the increase in
lower ozone values can be attributed to the increase in Asian emissions from 1980-1995 (Fiore et
al., 2002). The anthropogenic emissions and natural variability were found to have competing
effects on surface ozone over much of the US over 1980-2005 (Pozzoli et al., 2011). Another study
reproduced negative summertime ozone trends over the eastern US but underestimated the positive
trends in the western US likely due to underestimation of Asian emission trends or trans-pacific
transport or changes in stratosphere-troposphere exchange (Koumoutsaris and Bey, 2012). While
global models captured most of the observed variability and trends in summertime ozone, the use
of high-resolution regional models is recommended to reproduce interannual variability in winter
and spring in the western US (Strode et al., 2015).



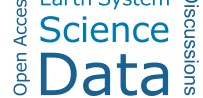

Apart from the interpretation of observed trends, the CTMs also provide information in
areas with no observations. However, CTM simulations suffer from both systematic (i.e., biases)
and random errors due to a number of factors, including numerical approximations, inadequate
understanding of some processes that control the spatial and temporal distribution of air pollutants,
inaccuracies in the initialization of the physical and chemical atmospheric state, and uncertainties
in the emission inventories. While continuous efforts are being made to improve the representation
of processes controlling $PM_{2.5}$ and ozone (Appel et al., 2010, 2013, 2017; Nolte et al., 2015; Fahey
et al., 2017) and emission inventories are updated by the EPA every three years, recent
developments have shown that assimilation of the National Aeronautics and Space Administration
(NASA) satellite retrievals of atmospheric composition in CTMs can significantly improve air
quality simulations (Gaubert et al., 2016; Kumar et al., 2019; Liu et al., 2011; Pagowski et al.,
2014; Saide et al., 2013). NASA satellite retrievals of atmospheric constituents with a far greater
spatial coverage compared to ground-based monitoring networks presents a unique opportunity to
develop long-term high-resolution air quality reanalysis, which can be useful for quantifying air
quality changes in unmonitored areas and assessing the impacts of changes in air quality on human
health and ecosystems.
This paper describes the methodology and evaluation of a long-term high-resolution air
quality reanalysis generated over the CONUS from 2005 to 2018 by assimilating the Moderate
Resolution Imaging Spectroradiometer (MODIS) aerosol optical depth (AOD) and the
Measurement of Pollution in the Troposphere (MOPITT) carbon monoxide (CO) retrievals daily
in the Community Multiscale Air Quality (CMAQ) model. An air quality dashboard developed to
enable the use of this dataset by a variety of stakeholders is also described.



## 2. Methodology

### 2.1. The Chemical Transport Model

The CMAQ model version 5.3.2 driven offline by the Weather Research and Forecasting (WRF) model version 4.1 is used to simulate air quality over the CONUS from 01 Jan 2005 to 31 Dec 2018. We employ the "cb6r3_ae7_aq" chemical mechanism that uses Carbon Bond 6 version r3 for gas-phase chemistry and AERO7 aerosol module for representing aerosol processes, including secondary organic aerosols (Appel et al., 2021). Both the WRF and CMAQ models use a horizontal grid spacing of 12 x 12 km$^2$ with WRF (CMAQ) grid using 481 (442), 369 (265), 36 (35) grid points in the longitudinal, latitudinal, and vertical directions, respectively. The model top is set to 50 hPa for both the models. The meteorological initial and boundary conditions for WRF are based on the six hourly ERA-Interim analyses at a grid spacing of 0.7° x 0.7°. We follow Appel et al. (2017) for physical parameterizations, four-dimensional data assimilation, and soil moisture nudging settings in WRF. We generate meteorology-dependent anthropogenic emissions for the EPA National Emissions Inventory (NEI) base years of 2011, 2014, and 2017 by feeding the WRF meteorological fields to the Sparse Matrix Operator Kernel Emissions (SMOKE). The emissions for 2005-2010 are derived by applying EPA reported annual state-wise trends to the NEIv2 2011 emissions. While NEI emissions are available for 2005 and 2008, the emissions processing platform for 2005 and 2008 does not process emissions for the "cb6r3_ae7_aq" chemical mechanism of CMAQ used here. Similarly, NEIv2 2014 emissions are used to derive emissions for 2012 and 2013, and the NEIv1 2017 emissions are used to derive anthropogenic emissions for the rest of the years. Fire emissions in CMAQ are represented using the Fire Inventory from NCAR (FINN) version 2.2 which provides daily varying global fire emissions at 1 x 1 km$^2$ resolution (Wiedinmyer et al., 2023). FINN emissions are processed through SMOKE to enable inline plume



rise of fire emissions within CMAQ. Biogenic emissions are calculated online within the model
using the Biogenic Emission Inventory System (BEIS). The chemical boundary conditions are
based on 6-hourly Whole Atmosphere Community Climate Model (WACCM) simulations (Marsh
et al., 2013; Gettelman et al., 2019). The WACCM output is mapped onto CMAQ grids using the
Initial Conditions Processor (ICON) and Boundary Conditions Processor (BCON).

**2.2.    Data Assimilation System**
We have used the three-dimensional variational (3DVAR) capability of the community Gridpoint
Statistical Interpolation (GSI) version 3.5 to assimilate the Level 2 MODIS AOD retrievals and
the Level 2 MOPITT CO retrievals in CMAQ. The MODIS AOD assimilation framework is the
same as we developed previously (Kumar et al., 2019) and the MOPITT CO assimilation capability
has been developed in this work. We use total aerosol mass per mode (Aiken, Accumulation, and
Coarse) and CO mixing ratios as the control variables in GSI. The state variables include individual
aerosol components, total aerosol mass per mode, CO mixing ratios, meteorological variables
(temperature, pressure, and relative humidity), and CMAQ vertical grid. Daily MODIS and
MOPITT retrievals are converted into a format compatible with GSI input modules.

A climatological background error covariance (BEC) matrix is generated separately for

winter (January) and summer (July) conditions using the GEN_BE tool, which reads two different
WRF-CMAQ runs driven by different meteorological and emission inputs but valid at the satellite
overpass time. Since there are multiple overpasses of the Terra and Aqua satellites that host the
MOPITT and MODIS sensors, we calculate the BEC at 15 Z, 18 Z, and 21 Z. The winter BEC is
used when assimilating satellite retrievals from November through March and the summer BEC is
used for the rest of the months. Our BEC design considers the uncertainties in meteorology,



anthropogenic, and biomass burning emissions. Meteorological uncertainties are represented by
using two different sets of physical parameterizations (Table A3.1) in two WRF runs to capture
errors in meteorology related to assumptions used in physical parameterizations. Species-
dependent perturbation factors for anthropogenic and biomass burning emissions are estimated by
comparing a number of available global/regional anthropogenic and biomass burning emission
inventories over the CONUS (Table A3.2 and A3.3). Among the two WRF-CMAQ runs fed to
GEN_BE for BEC estimation, we used the default emissions in the first run and perturbed the
emissions in the second run. The BEC was then estimated in terms of variances and length scales
(both horizontal and vertical) for total aerosol mass per mode and CO, and used in GSI. We refer
the reader to Kumar et al. (2019) for a description of BEC parameters.

We have assimilated standard Level 2 Collection 6.1 MODIS AOD and Version 8

MOPITT CO retrievals based on the multispectral algorithm (thermal and near infrared) in CMAQ.
This multispectral product is more sensitive to near-surface CO over land compared to the thermal-
infrared only retrievals. MOPITT retrievals agree with in-situ measurements at all vertical levels
within ±5% (Deeter et al., 2019). The observation errors for MODIS AOD retrievals are specified
as (0.03 + 0.05 * AOD) and (0.05 + 0.15 * AOD) over the ocean and the land, respectively (Remer
et al., 2005). The observation errors for CO profiles are used as reported in the MOPITT retrieval
product. A simple forward operator and its adjoint based on the parameterization of (Malm and
Hand, 2007) is used to convert CMAQ aerosol chemical composition into AOD for a direct
comparison with MODIS AOD retrievals as described in Kumar et al. (2019). The forward
operator and its adjoint for MOPITT CO assimilation are developed in this study and described in
Appendix A1.



## 2.3. Reanalysis production workflow

Daily analyses of three-dimensional fields of aerosols and CO based on the assimilation of MODIS AOD and MOPITT CO retrievals in CMAQ using the GSI system has been performed using the workflow shown in Figure 1. The first CMAQ simulation on 01 Jan 2005 is initialized using the global model simulations from WACCM, and all subsequent simulations until 31 Dec 2018 are initialized from the previous CMAQ simulations. Every day, we perform 9 simulations, with the first simulation running CMAQ from 00-15 Z, the second simulation assimilating MODIS Terra and Aqua AOD retrievals at 15 Z, and third simulation assimilating MOPITT CO retrievals at 15 Z. The fourth simulation advances CMAQ from 15 Z to 18 Z with the fifth and sixth simulations assimilating MODIS AOD and MOPITT CO at 18Z, respectively. The seventh simulation advances CMAQ from 18 Z to 21 Z, the eighth simulation assimilates MODIS Aqua AOD retrievals at 21 Z, and the ninth simulation advances CMAQ from 21 Z to 00 Z of the next day. This resulted in a total of 46,152 jobs submission on the NCAR supercomputer Cheyenne (https://arc.ucar.edu/knowledge_base/70549542). An automated script was developed to submit and track successful completion of these jobs.

The assimilation times of 15 Z, 18 Z, and 21 Z were determined based on the analysis of overpass times of Terra and Aqua satellites, which pass over the CONUS between 13:30 Z and 22:30 Z. All the satellite retrievals belonging to a 3-hour window are assumed to be available for assimilation at the center of that window. For example, all the satellite retrievals between 1330 Z and 1630 Z are assimilated at 1500 Z.

Our previous work has shown that the assimilation of MODIS AOD in CMAQ improved the correlation coefficient between CMAQ simulated and independently observed $PM_{2.5}$ by ~67% and reduced the mean bias by ~38% over the CONUS during July 2014. To understand whether





GSI pushes CMAQ towards MOPITT, we performed and compared one month (July 2018) of
CMAQ experiments with and without assimilation of MOPITT CO profiles. We find that the
assimilation of MOPITT CO profiles substantially improves the correlation coefficient and reduces
the errors (both mean bias and root mean squared error) between CMAQ and MOPITT CO at all
the pressure levels except at 100 hPa where the MOPITT sensitivity is the lowest (Appendix A2,
Figure A2.1). This simple test confirms the ability of GSI to constrain the performance of CMAQ
with satellite observations. Other trace gas species (e.g., ozone and OH) are not affected directly
by the assimilation of AOD and CO, but the impact of assimilation indirectly affects these species
through photochemical processes in the model.

**2.4.    Output frequency and optimization**
The production of a chemical reanalysis also poses a challenge of storing the model output. Since
our chemical reanalysis focuses on air quality applications, we saved all the chemical variables
together with relevant meteorological parameters (2 m temperature and relative humidity, 10 m
wind speed and direction, planetary boundary layer height, precipitation, and downward reaching
solar radiation) and deposition (both dry and wet) fluxes every hour at the surface. The total size
of this output is 12 Terabytes.

**3.    Ground-based observations and trend calculation method**
We have obtained and processed hourly in-situ measurements of 2 m temperature (T2), 2 m relative
humidity (RH2), 10 m wind speed (WS10), 10 m wind direction (WD10), and surface pressure
from the METeorological Aerodrome Reports (METAR) network, which is distributed by the
NCEP's Meteorological Assimilation Data Ingest System (MADIS). METAR data are surface



weather observations and it consists of meteorological data from airports (Automated Surface
Observing Systems) and other permanent weather stations (Automated Weather Observing
System) located throughout the US. We used the Level-3 Quality Controlled METRAR data over
CONUS     to     evaluate     our     modeled     meteorological     fields
(https://madis.ncep.noaa.gov/madis_metar.shtml). Daily precipitation data from the 0.1-deg
Integrated Multi-satellitE Retrievals for Global precipitation measurements (IMERG;
https://gpm.nasa.gov/data/imerg) dataset is used to evaluate WRF simulated precipitation.

To evaluate the modeled surface $PM_{2.5}$ and ozone concentrations, we have obtained hourly

$PM_{2.5}$ and ozone observations from the EPA Air Quality System, which currently measures $PM_{2.5}$
and ozone at more than 1000 sites across the US. The AQS data also contains values below the
method detection limit (MDL). The MDLs are different for ozone and $PM_{2.5}$ and also vary as a
function of site and instrument type. For consistency, we assume the MDL values of 5 ppb for
ozone and 2 $\mu g/m^3$ for $PM_{2.5}$ for all sites. All the data below MDL was replaced by MDL/2
(https://www3.epa.gov/ttnamti1/files/ambient/airtox/workbook/AirtoxWkbk4Preparingdataforan
alysis.pdf; https://pubs.acs.org/doi/10.1021/es071301c). The sites for which two simultaneous
measurements (corresponding to two instruments) were available, the mean value is taken for
further calculation.

The trend calculations were performed using both the observed and modeled ozone and

$PM_{2.5}$ values. The monthly mean time series of observed and modeled maximum daily 8-hour
(MDA8) ozone and 24-hour average $PM_{2.5}$ during 2005-2018 is calculated over all measurement
sites. The daily MDA8 ozone over a site is calculated using the EPA's defined methodology
(https://www.govinfo.gov/content/pkg/FR-2015-10-26/pdf/2015-26594.pdf, pp 168). For each
day, 8-hour running averages are taken from 7 am to 11 pm local standard time, which constitutes





17 8-hour running mean values per day. If an 8-hour window has less than 6 hours of data and the
mean value of the remaining hours is less than 70 ppb then the data for that window is discarded.
If a site has fewer than 13 valid 8-hour mean values or the maximum value of the available 8-hour
average is less than 70 ppb then the value for that day is discarded. For $PM_{2.5}$, a daily 24-hour
average value is calculated in local standard time only if at least 18 hours of valid data/day are
available. Furthermore, we discarded all sites with (1) < 50% data per month, (2) < 50 % data
during each year, and (3) if number of years with ≥ 50% data were < 10 years during 2005-2018.
The number of valid sites fulfilling the above criteria over CONUS are estimated to be 1012 and
369, for MDA8 ozone and 24-hour $PM_{2.5}$, respectively. Daily values of MDA8 ozone and 24-hour
$PM_{2.5}$ are used to calculate monthly 5th percentile, 50th percentile, 95th percentile and mean time
series during 2005-18 at each valid site. A similar criterion for seasonal mean, 5th, 50th and 95th
percentile time series was also used. The number of valid sites during summer season were the
maximum (1010/357 for MDA8 $O_3$/24-hour $PM_{2.5}$) and were minimum (501/337 for MDA8
$O_3$/24-hour $PM_{2.5}$) during the winter season. These annual and seasonal MDA8 ozone and $PM_{2.5}$
time series are then used to estimate annual and seasonal trends and the significance of trend values
are also tested.

**4.    Results and Discussions**
**4.1.    Meteorological evaluation**
The WRF simulations for the entire period (2005-2018) processed using the Meteorology-
Chemistry Interface Processor (MCIP) are collocated with METAR observations of T2, RH2,
WS10, and WD10 in space and time, and paired values are used for evaluating the model. The
evaluation is performed at a regional scale following the EPA regional classification of the





CONUS in 10 regions. The number of METAR sites during 2005-2018 was 1290, and the
maximum available hourly data during the study period was 33-68 % over 10 EPA regions. Region
8 has the least data (~33-37%), and other regions have 47-68 % data during 2005-2018. Hourly
regional averaged model and METAR observations time series are compared over 10 EPA regions
for T2 (Figure 2), RH2 (Figure 3), WS10 (Figure 4), and WD10 (Figure 5). Three statistical
metrics, namely correlation coefficient (r), mean bias (MB), and root mean square error (RMSE),
for each region are also listed in Figures 2-5.

Hourly regional averaged T2 between model and observations (Figure 2) show excellent

correlations of 0.8-1.0 with low mean biases of -0.3 to 0.4 °C and the RMSE ranging from 2.0-5.7
°C over the 10 EPA regions. The model also performed well (r = 0.7-0.9) in simulating RH2
(Figure 3) over 10 EPA regions with the mean biases of 0.9-3.6 %.and the RMSE of 12.5 - 16.3
%. Since RH2 is estimated as a ratio of vapor pressure to saturation pressure (es) and es depends
on T2, the biases in T2 also contribute to the biases in RH2. For example, EPA Region 6 which
shows the highest T2 RMSE also shows the highest RH2 RMSE. The model reproduces the
variations in surface pressure very well (r = 1.0) with a slight underestimation (MB = -8.1 to 0.2
hPa; RMSE = 0.3-8.1 hPa). The slight underestimation in pressure is seen in eight out of 10 EPA
regions with the largest MB in Regions 9 (-8.1 hPa) and 10 (-7.4 hPa). The errors in surface
pressure (plot not shown) over these regions could also contribute to biases in T2 and RH2.

Prior to 10 m wind speed comparison, model wind speeds are assigned "zero value" if the

hourly wind speed at any site is less than 0.51 m/s (1 knot). This step was needed to make model
output consistent with the METAR wind speed data, which treats such wind speeds as calm winds
and assigns it a zero value. Our model simulation slightly overestimates (MB = 0.1-0.8 m/s) WS10
(Figure 4) over most of EPA regions with the exception of Region 8 (MB = -0.1 m/s). Wind



direction (Figure 5) biases (absolute) over these regions were 34º-58º. The correlation coefficients
for both WS10 and WD10 are slightly lower in Regions 8-10, which is likely due to the complex
topography in these regions. The correlation coefficients for 10 m wind speed were lower than
those for temperature, and relative humidity, indicating a slightly poorer model performance for
winds. The WRF model is known to overpredict 10 m wind speed at low to moderate wind speeds
in all available planetary boundary layer (PBL) schemes (Mass and Ovens, 2010). This
shortcoming of the model was partly attributed to unresolved topographical features by the default
surface drag parameterization, which in turn influences surface drag and friction velocity, and
partly to the use of coarse horizontal and vertical resolutions of the domain (Cheng et al., 2005).

Since WRF and IMERG precipitation have different resolutions, we first mapped the WRF

simulated precipitation from a 12 km x 12 km grid on Lambert conformal projection to the IMERG
rectilinear grid of 0.1º x 0.1º using the "rcm2rgrid" functionality of the NCAR command language
(https://www.ncl.ucar.edu/Document/Functions/Built-in/rcm2rgrid.shtml). The seasonal mean
WRF simulated and IMERG derived precipitation are then compared over four seasons during
2005-2018 (Figure 6). The model is able to capture the spatial patterns in precipitation in different
seasons, with an underestimation of -0.1 to -0.9 mm/day. The highest underestimation is observed
during the winter season. The eastern CONUS showed an underestimation during winter, spring
and autumn seasons, however, over the western US, the model mostly overestimated the
precipitation, especially in the mountainous regions (Rockies, Cascades, and Sierra Nevada). The
model also showed larger biases over the lakes and oceanic regions. Despite the biases, this
comprehensive evaluation shows that our model simulations captured the key features of regional
and temporal variability of the key meteorological parameters over the CONUS fairly well.



## 4.2. Air Quality evaluation

Hourly regional averaged observed and CMAQ simulated surface ozone and $PM_{2.5}$ are compared for all the EPA regions in Figures 7 and 8, respectively. In all the regions, the model captures the seasonal cycle in surface ozone characterized by a summertime peak as well as the observed interannual variability very well, with correlation coefficients of 0.77 to 0.91. The model also overestimates the nighttime ozone levels in all the regions, but a larger overestimation is seen in Regions 8 and 9. The mean bias and RMSE in modeled ozone are very similar across the regions, with values ranging from 3.7 - 6.8 ppbv and 7.0-9.0 ppbv, respectively. The model shows a slightly poorer skill in capturing the variability in $PM_{2.5}$ relative to ozone as reflected by smaller r values of 0.49-0.79 but captures long-term trends in most of the regions reasonably well. The mean bias and RMSE in modeled $PM_{2.5}$ are estimated to be -0.9 to 5.6 µg/m$^3$ and 3.0 to 8.3 µg/m$^3$, respectively. The largest underestimation of $PM_{2.5}$ is seen in Region 8, particularly from 2005 to 2012 while the largest overestimation is seen in Region 2.

In addition to regional evaluation, we also evaluated the model performance for different land use types and location settings (see Appendix A2, Figure A2.2 for classification of the number of sites in these categories). This categorization information by land use and location types was not available for a very small number of sites, and thus, they were excluded from the analysis (sites classified as "NONE" in Figure A2.2). Since Maximum Daily Averaged 8-hour (MDA8) ozone and daily averaged $PM_{2.5}$ are policy-relevant metrics, we focus on the evaluation of these parameters on a monthly averaged scale for this evaluation. We evaluate monthly median (50$^{th}$ percentile), 5$^{th}$ and 95$^{th}$ percentile time series of MDA8 ozone, and daily averaged $PM_{2.5}$ for different land use categories and location settings (Appendix A2, Figures A2.3-A2.8).



Among the rural sites, all land use categories showed the highest biases for the 5th
percentile, followed by the median and 95th percentile for MDA8 ozone, except for the "Others"
category, for which the median showed the lowest bias. For suburban and urban site types, 95th
percentile MDA8 ozone consistently showed the lowest bias for all land use types, followed by
the median and 5th percentile. Furthermore, "Others" land use category under the rural and urban
sites shows the lowest bias for 5th percentile and the median, while "residential" land use type
shows the lowest bias for the suburban sites.
For $PM_{2.5}$, the largest differences between the model and observations are seen for the 95th
percentile at "Others" land use categories compared to the 5th percentile and median. The model
generally captures the temporal variability in $PM_{2.5}$ across all land use types (except "Others") and
location settings for all three-percentile metrics analyzed here but some anomalies are also evident.
For example, residential and commercial sites in the urban category show larger overestimation
for the median and 95th percentiles during 2005-2006, indicating higher uncertainties in
anthropogenic emission estimates at these sites during these years. While the model follows most
of the observed peaks in 95th percentile, it substantially underestimates the observed peaks.
The errors in air quality simulations can be attributed to the uncertainties in different types
of emissions used to drive air quality models, errors in the lateral boundary conditions representing
pollution inflow, uncertainties in meteorological parameters (as quantified earlier in this section),
and poor understanding of some of the physical and chemical processes controlling the fate of
those emissions. To quantify uncertainties in anthropogenic and biomass burning emissions over
the CONUS, we compared all available anthropogenic and biomass burning emission inventories
over the CONUS and found that anthropogenic emission estimates across various emission
inventories vary by a factor of 1.16 - 2.94 (Table A3.2) and the corresponding fire emission



estimates vary by 3.13 - 8.0 (Table A3.3). The extrapolation of the NEI emissions to years other
than the base years might have also introduced some uncertainties in our simulations because EPA
reported state level trends may not always represent local (sub-state) changes in emissions and
also do not provide information about new emission sources appearing in the CONUS between
two NEI base emission inventory years.

**4.3.     Trend analysis**
The spatial distribution of positive/negative trend values in MDA8 ozone and 24-hr average $PM_{2.5}$
calculated using monthly median values in AQS and CMAQ data during 2005-2018 are shown in
Figures 9 and 10, respectively. Different symbols are used to represent urban, suburban, and rural
site types. Based on location, ~42/23% of sites were in rural areas, ~41/45 % in suburban areas
and ~17/32% were in urban or city centers, respectively, for MDA8 ozone/24-hr average $PM_{2.5}$.
Darker/lighter red and blue colors represent statistically significant/insignificant increasing and
decreasing trends at 2-sigma level. Over the study period, both the model and observations show
decreasing trends in MDA8 ozone over the majority of the CONUS. Most eastern US sites show
decreasing trends that were statistically significant with p values less than 0.05. The sites located
in western/northwestern US, however, showed mixed results with some sites showing increasing
trends, most of which were not statistically significant. Similar results were observed during the
summer season with most sites showing statistically significant decreasing trends over the most
locations. During autumn and winter seasons, several sites over California and the eastern US
showed decreasing but insignificant trends. Some sites over the midwestern US also changed the
trend sign during these seasons. The trends in winter seasons were mostly positive (increasing,
55% agreement between AQS and CMAQ) over most sites in the US (except for the coastal sites



in the southeastern US). The number of sites with negative trend values in summer changed from
3% to 55% positive trends during winter season. Most of these trends, however, were not
statistically significant and the number of sites was also reduced to about 560% of the sites
available for annual evaluation during this season. The seasonal changes in monthly median trends
discussed above were mostly consistent (67-86%) between the AQS and CMAQ data. A similar
analysis with 5[th] and 95[th] percentile time series suggested that the higher percentiles showed mostly
decreasing trends, but 5[th] percentile dataset at the mid-western US, Boston-New York-DC, and
central US sites showed increasing trends on a seasonal and annual basis. The MDA8 ozone trend
over CONUS (1012 sites) is estimated to be -0.53 ± 0.46/-0.56 ± 0.45 ppb/year (summertime) and
-0.31 ± 0.43/-0.29 ± 0.39 ppb/year (annual), respectively, for AQS/CMAQ data, with most sites
(~70 %) showing negative trends. At the 2-sigma level (p-value < 0.05), the summertime mean
ozone trends are -0.85± 0.36/-0.75 ± 0.35 ppb/year for 484/620 sites and annual MDA8 ozone
trends are -0.52 ± 0.45/-0.47 ± 0.42 ppb/year for 554/562 sites, respectively, for AQS/CMAQ data
over CONUS. This suggests decreases in monthly high ozone days but increases in monthly low
ozone. On an annual basis, MDA8 ozone showed the most decreasing trends (AQS/CMAQ= -0.40
± 0.37/-0.34 ± 0.34 ppb/year) in the 428 rural sites. The mean ozone trends over urban (411 sites)
and suburban (170) areas were (AQS/CMAQ = -0.28 ± 0.44/-0.29 ± 0.40 ppb/year) and
(AQS/CMAQ = -0.13 ± 0.48/-0.15 ± 0.48 ppb/year), respectively. The ozone trends over high-
altitude sites (16 sites), are mostly negative for AQS/CMAQ = -0.43 ± 0.45/-0.12 ± 0.36 ppb/year)
in summer and annually (AQS/CMAQ, = -0.39 ± 0.38/-0.03 ± 0.29 ppb/year).

Similar MDA8 ozone trends were also reported in a previous study (Simon et al., 2015).

Mousavinezhad et al. (2023) reported that all regions except the Northern Rockies and the
Southwest experienced decreasing trends in median MDA8 ozone values during the warm season



of 1991-2020, with rural stations in the Southeast and urban stations in the Northeast experiencing
the greatest declines of -1.29 ± 0.07 ppb/year and -0.85 ± 0.08 ppb/year, respectively. They also
reported a large decrease in MDA8 ozone 95th percentile in all regions. Similarities in ozone trends
between the AQS observations and CMAQ simulations over a longer time period 1990-2015 is
also reported by He et al. (2020).

On an annual basis, 24-hr average $PM_{2.5}$ also showed mostly decreasing trends (~79 %)

over most of the sites. A majority of these trends were also statistically significant at 2-sigma level
(AQS/CMAQ = 70 %/75 %). However, unlike MDA8 ozone, an increasing trend (though
insignificant) in summertime $PM_{2.5}$ is observed over the north-western US (Fig. 9). The wintertime
trends were also mostly decreasing over most of the sites, except for the northwestern US. During
summer season about 5-fold increase (annual ~ 5%; summer ~ 24%) in positive trends is observed
in high $PM_{2.5}$ days (95th percentile time series) and most of these increases were observed over the
Pacific Northwest. These summertime increases in $PM_{2.5}$ trends are also evident from the 95th
percentile time series, where a sharp increase in $PM_{2.5}$ is observed during 2017-2018 overall sites
except industrial locations (refer to Fig. S8). In recent years these changes could be even stronger
as wildfire activity over the western US has increased in the last decade. The dramatic decreasing
trends of $PM_{2.5}$ in the eastern US were also reported in previous studies (Zhang et al., 2018; Gan
et al., 2015; Xing et al., 2015) (Gan et al., 2015; Xing et al., 2015; Zhang et al., 2018) due to
emission reductions. The increasing trend in the western central area is due in part to frequent
wildfires (Dennison et al., 2014; McClure and Jaffe, 2018). For $PM_{2.5}$ the overall mean trends are
-0.24 ± 0.21/-0.24 ± 0.24 μg/m³/year (369 sites) in AQS/CMAQ data sets. Unlike, MDA8 ozone,
the number of sites remained almost the same (337-357 sites in four seasons, 369 annual) during
seasons and an overall negative trend is also observed (-0.18 ± 0.25 to -0.30 ± 0.35 μg/m³/year).

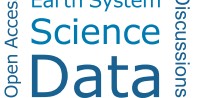

At 2-sigma level, the number of sites that showed negative trends in both the datasets were 69-80

436 %.

On an annual basis, the mean $PM_{2.5}$ trends over urban sites are -0.17 ± 0.22/-0.18 ± 0.15
$\mu g/m^3$/year, suburban sites are -0.28 ± 0.22/-0.24 ± 0.26 $\mu g/m^3$/year and -0.3 $\mu g/m^3$/year, and urban
and city center are -0.23 ± 0.21/-0.30 ± 0.27 $\mu g/m^3$/year $\mu g/m^3$/year, respectively, for AQS/CMAQ
data. The only high-altitude site for $PM_{2.5}$ showed an increase in the annual (0.07/0.06 $\mu g/m^3$/year
for AQS/CMAQ data) and summertime trend (0.13/0.13 $\mu g/m^3$/year for AQS/CAMQ data).
During other seasons, mostly low negative trends were observed. The ozone trends over high-
altitude sites (16 sites), however, are mostly negative (-0.43 ± 0.45/-0.12 ± 0.36 ppb/year in
summer and -0.39 ± 0.38/-0.03 ± 0.29 ppb/year, annually). The ozone trends at high altitude sites
showed large seasonal variations with min to max ranges of -0.69 to 0.87/-1.5 to 0.26 ppb/year for
AQS/CMAQ data.

**4.4.    Air Quality dashboard**
The comprehensive evaluation of our reanalysis in the above sections shows that our
reanalysis is able to capture key features of long-term trends in both MDA8 ozone and $PM_{2.5}$ over
most parts of the CONUS. This increases confidence in using this dataset for assessing the trends
in unmonitored areas of the CONUS. Therefore, a Geographic Information System (GIS)-based
dashboard has been developed to aid in community engagement and understanding of the
reanalysis data. The dashboard was developed using Esri ArcGIS Dashboard technology (Esri,
2024). An interactive web-based dashboard allows stakeholders to explore air quality annual
concentrations and the number of days that exceed a certain threshold over space and time. It
provides a step-by-step path for users to explore information at the CONUS, state, and county



levels. In the center of the dashboard is a time series chart showing trends in annual concentrations
of MDA8 ozone, NO2, PM$_{2.5}$, PM$_1$, and PM$_{10}$ between 2005 and 2018. An indicator element of a
dashboard highlights how many days between 2005 and 2018 have exceeded the National Ambient
Air Quality Standards (NAAQS) for ozone and PM$_{2.5}$, and a bar chart graph shows the number of
days that exceeded the NAAQS each year. There is also a map that zooms to the selected state or
county of interest and illustrates the spatial distribution of air quality variables using a quantitative
color bar.

The dashboard can be used to better understand how particular events, such as large

wildfires, have affected air quality in certain geographic areas. For example, the 2008 wildfires in
Shasta and Trinity Counties in California, referred to as the June Fire Siege, had a major impact
on air quality (https://storymaps.arcgis.com/stories/c6535ee477e14b72a20393a5f10aefbc). Figure
11 shows MDA8 ozone concentrations for Shasta County, California. The dashboard shows a
sharp increase in MDA8 ozone concentration in 2008, as depicted in the time series plot.  The bar
chart in the lower right corner also reflects the large number of days that exceeded the NAAQS
criteria for MDA8 ozone in 2008.

The dashboards also can be used to visualize the efficacy of air quality management

policies. For example, Los Angeles County, CA has designed and implemented strict emission
standards to improve air quality. Figure 12 shows the downward trend in PM$_{2.5}$ concentrations in
Los Angeles County during 2005-2018. The air quality dashboard is publicly accessible at
https://ncar.maps.arcgis.com/apps/dashboards/9a97650dc77b4f7192b99ea9bef36a21.

We have also developed a Python-based Streamlit application allowing users to select and

download data for specific time periods aggregated over administrative boundaries such as cities,
counties, and states. Temporal and spatial aggregations are performed on the server, and only



information of interest is downloaded and delivered to the users, taking the data processing
workload off of the users. The Streamlit application allows users to select a time period, a temporal
aggregation (daily, weekly, monthly, annual), one or more air quality variables, statistics (min,
mean, max), and an area of interest (state, county, city). The data can then be downloaded as a
comma-separated file as well as graphed on the website as seen in Figure 13. The Streamlit
application is available at: https://compass.rap.ucar.edu/airquality/

**5. Data availability**

The global meteorological datasets used to drive WRF are publicly available through National
Center for Atmospheric Research (NCAR) Research Data Archive (https://rda.ucar.edu/). The
SMOKE setup used to create emissions for CMAQ is accessible via EPA emissions modeling
platform (https://www.epa.gov/air-emissions-modeling/emissions-modeling-platforms). FINN
biomass burning emissions can be downloaded from https://rda.ucar.edu/datasets/ds312.9/.
Meteorological observations used to evaluate the model performance are downloaded from
https://madis-data.cprk.ncep.noaa.gov/madisPublic1/data/archive/. The EPA AQS system
observations are downloaded from https://www.epa.gov/aqs. Hourly surface output from the
WRF-CMAQ-GSI system can be downloaded from https://doi.org/10.5065/cfya-4g50 (Kumar and
He, 2023)



**6. Code availability**



The WRF, CMAQ, and GSI source codes are publicly accessible at https://github.com/wrf-model/,
https://github.com/USEPA/CMAQ,        and        https://dtcenter.org/community-code/gridpoint-
statistical-interpolation-gsi/download.

**7.    Conclusions**

Air pollution is an important health hazard affecting human health and the economy in the CONUS
but still, millions of people in counties with no monitors. To address this gap and help air quality
managers understand long-term changes in air qualities at the county level across the CONUS, we
have created a 14-year long 12-km hourly dataset by daily assimilation of atmospheric composition
observations from the NASA MODIS and MOPITT sensors aboard the Terra and Aqua satellites
in the Community Multiscale Air Quality (CMAQ) model from 01 Jan 2005 to 31 Dec 2018.   The
WRF model has been used to simulate meteorological parameters, which are then used to drive
CMAQ offline and for generating meteorology-dependent anthropogenic emissions.

The meteorological parameters, ozone, and $PM_{2.5}$ have been extensively validated against

multi-platform observations to characterize uncertainties in our dataset, which air quality managers
need to determine the confidence they can put in our dataset. We show that our dataset captures
regional scale hourly, seasonal, and interannual variability in the meteorological variability well
across the CONUS. The model shows an excellent performance in simulating the regional and
temporal variability in temperature and relative humidity but a slightly poorer performance in
simulating winds and precipitation, which are well known shortcomings of the WRF model. The
model also shows a higher skill in reproducing variabilities in surface ozone (r = 0.77-0.91) than
$PM_{2.5}$ (0.49-0.79). The mean biases for CMAQ ozone and $PM_{2.5}$ are estimated to be 3.7-6.8 ppbv





and -0.9-5.6 µg/m$^3$, respectively, and the corresponding RMSE values are 7-9 ppbv and 3.0-8.3
µg/m$^3$ , respectively.
The MDA8 ozone trend over CONUS is estimated to be -0.53 ± 0.46/-0.56 ± 0.45 ppb/year
(summertime) and -0.31 ± 0.43/-0.29 ± 0.39 ppb/year (annual), respectively, for AQS/CMAQ data
with ~70% of sites showing negative trends. At a 2-sigma level, the summertime MDA8 ozone
trends are -0.85 ± 0.36/-0.75 ± 0.35 ppb/year and annual MDA8 ozone trends are -0.52 ± 0.45/-
0.47 ± 0.42 ppb/year, respectively, for AQS/CMAQ data over CONUS. Annually, at 2-sigma level,
46% sites showed negative trends in both the data. Annual mean PM$_{2.5}$ trends are –0.24 ± 0.21/-
0.24 ± 0.24 µg/m$^3$/year, respectively in AQS/CMAQ data sets, and ~79% of the sites showed
negative trends. Annually, at 2-sigma level, 66% sites showed negative trends in both the data.
During summertime, the negative trend percent is reduced to 71%, where an increase in positive
trends are observed in the northwestern US.
An air quality dashboard has been developed, which provides a step-by-step path for users
to explore information at the CONUS, state, and county levels. This dashboard allows the users to
visualize air quality information in the form of maps, bar charts, and the NAAQS exceedance days.
Finally, a Python-based Streamlit application is developed to allow the download of the air quality
data in simplified text and graphic formats for the end user's choice of the region and time of
interest.



**8.    Figures**

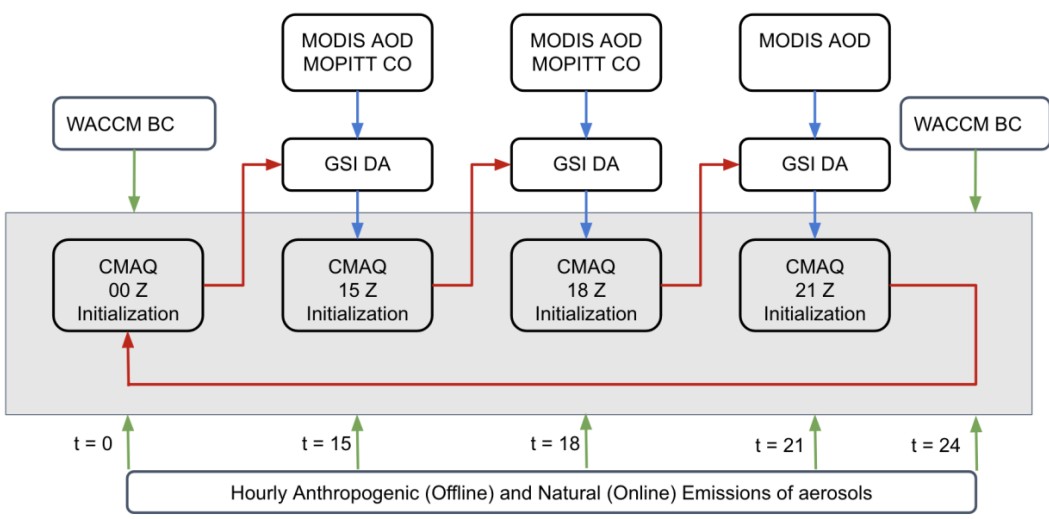


**Figure 1:** *Architecture of the daily GSI/CMAQ based chemical data assimilation workflow.*










**Figure 2:** Time series of hourly averaged 2 m temperature over 10 EPA regions (R1-R10) from

WRF-CMAQ setup (red) and METAR observations (black) during 2005-2018. Orange and Grey

lines represent the standard deviation for WRF-CMAQ and METAR, respectively. The correlation



coefficient (r), mean bias (MB), and the root mean square error (RMSE) for each region is also
shown.




**Figure 3:** Same as Figure 2 but for 2 m relative humidity.




**Figure 4:** Same as Figure 2 but for 10 m wind speed.







**Figure 5:** Time series of hourly averaged 2 m temperature over 10 EPA regions (R1-R10) from

WRF-CMAQ setup (red) and METAR observations (black) during 2005-2018. Orange and Grey

lines represent standard deviation for WRF-CMAQ and METAR, respectively. The correlation
coefficient (r), mean absolute bias (MAB), and the root mean square error (RMSE) for each region
is also shown.

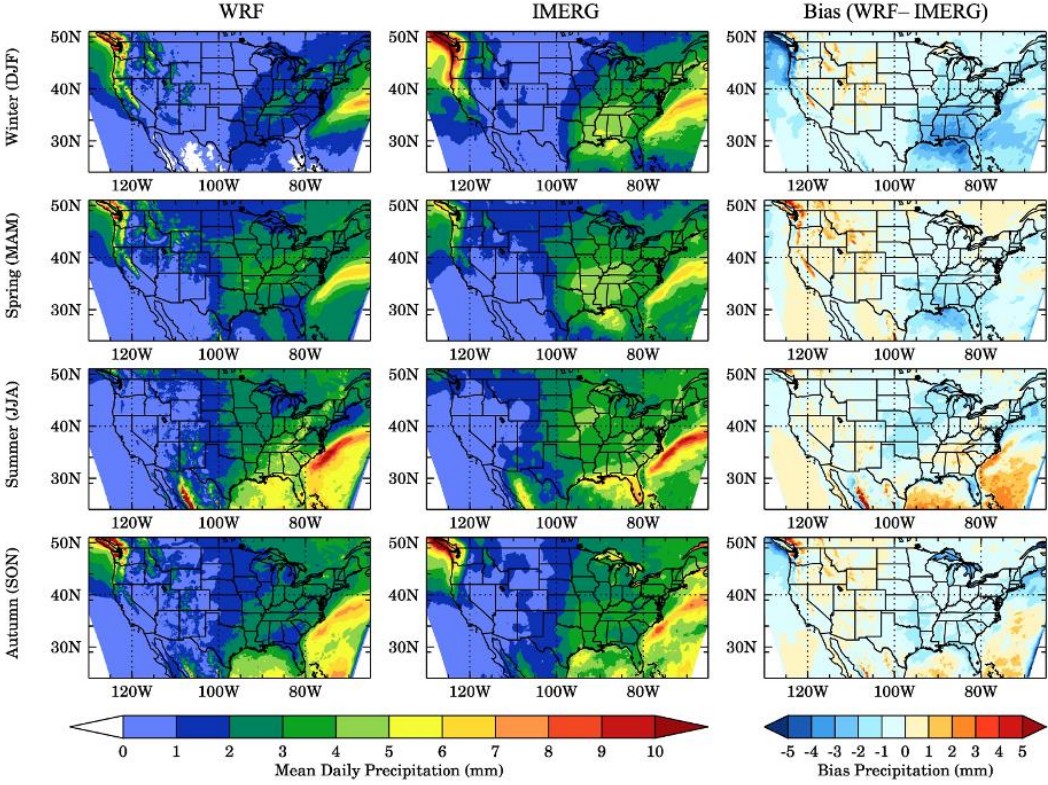


**Figure 6:** Spatial distribution of mean daily precipitation and bias during four seasons in 2005-
2018 (top to bottom, *viz.,* Winter, Spring, Summer and Autumn). Left, center and right panels
represent mean precipitation from WRF, IMERG and bias (WRF-IMERG) precipitation,
respectively.




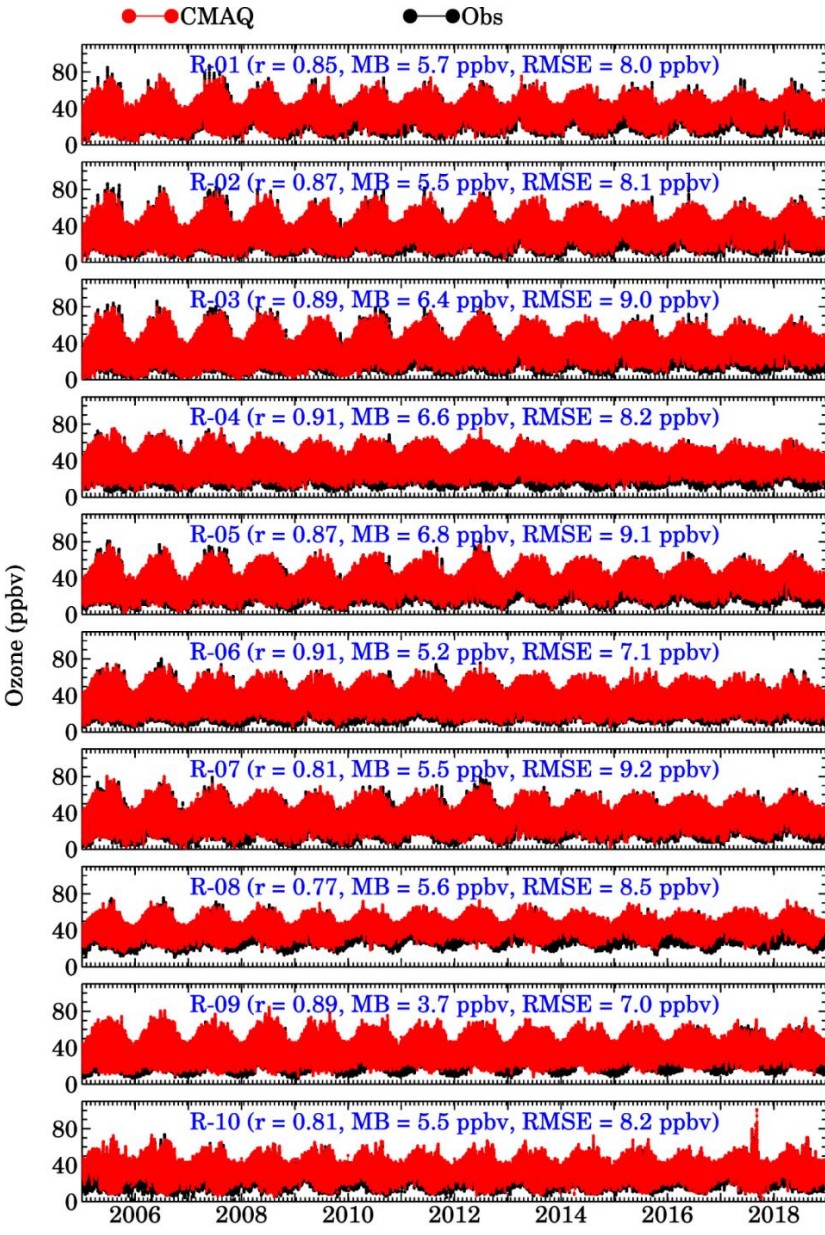


**Figure 7:** Time series of hourly averaged surface ozone over 10 EPA regions (R1-R10) from

WRF-CMAQ setup (red) and EPA AQS observations (black) during 2005-2018. The correlation

coefficient (r), mean bias (MB), and the root mean square error (RMSE) for each region is also
shown.

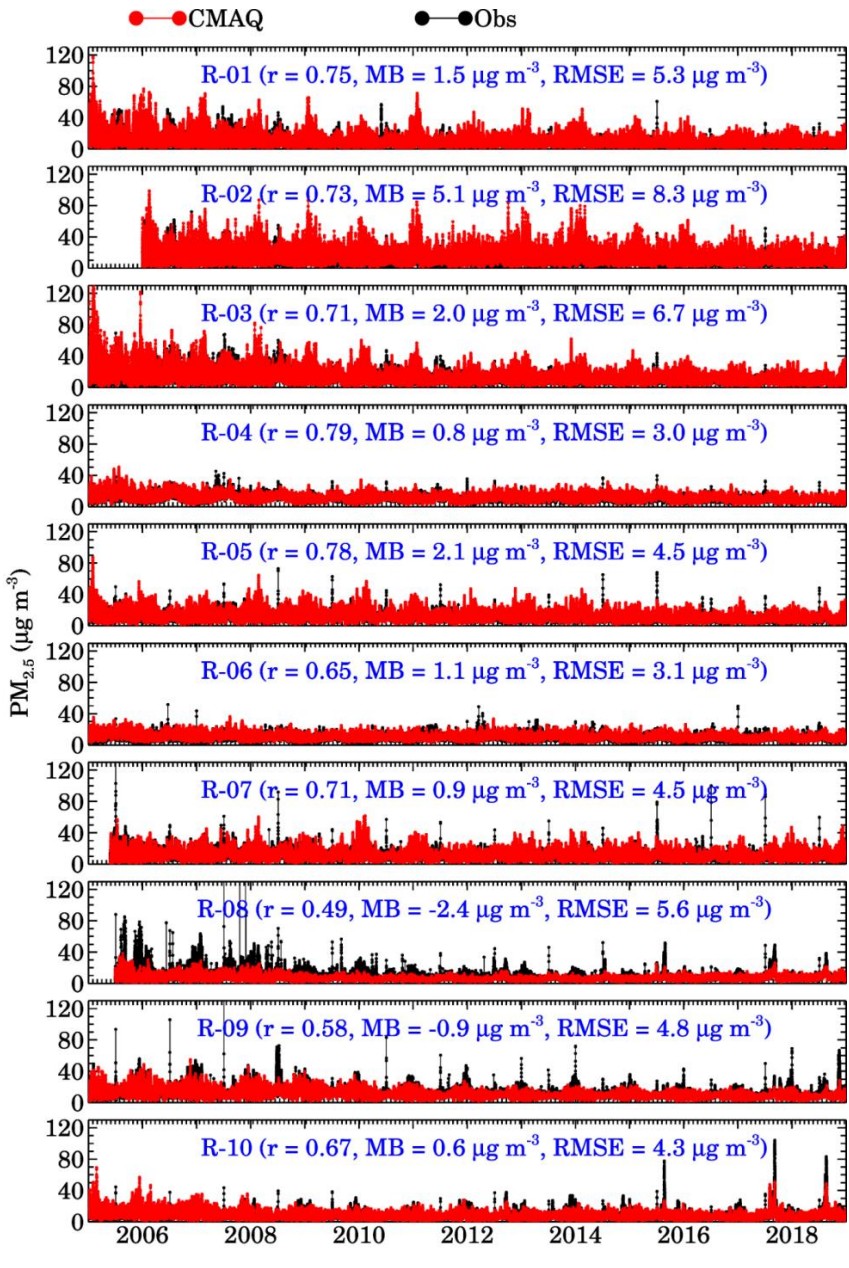


**Figure 8:** Same as Figure 7 but surface fine particulate matter (PM₂.₅).



Ozone [Monthly 50ᵗʰ percentile timeseries]

**Figure 9:** Spatial distribution of positive (blue colors), negative trends (red colors) in MDA8 ozone at different statistically significant levels (p-values) using annual, seasonal monthly median time series (top to bottom). Plots on the right show differences in trend values [CMAQ-AQS].




**Figure 10:** Spatial distribution of positive (blue colors), negative trends (red colors) in 24-hour

avg. PM$_{2.5}$ (right panel) at different statistically significant levels (p-values) using monthly median

time series (top to bottom). Plots on the right show differences in trend values [CMAQ-AQS].





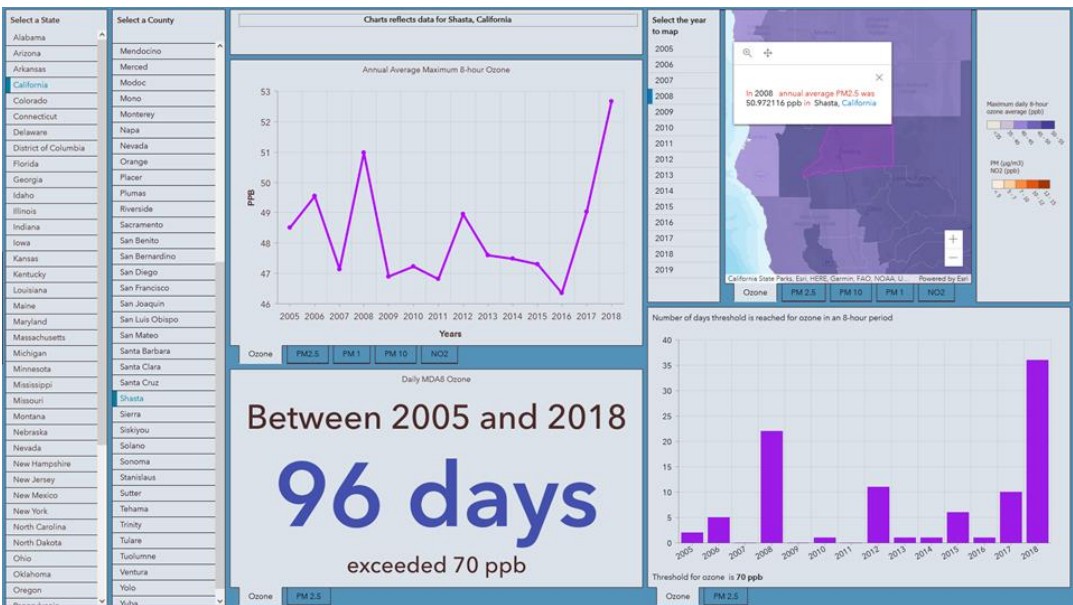

**Figure 11:** Dashboard reflecting Ozone concentrations for Shasta, CA.







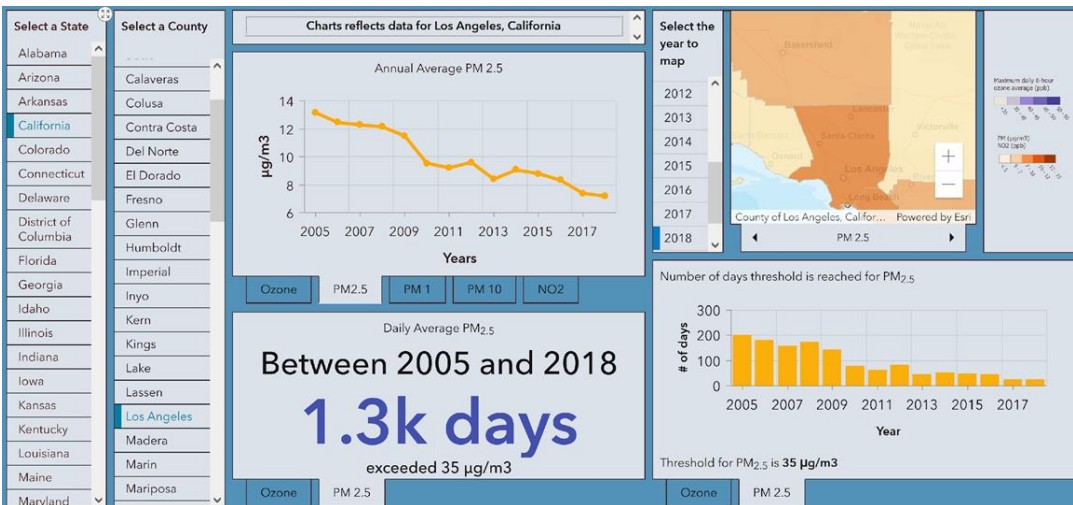

**Figure 12:** PM$_{2.5}$ concentrations for Los Angeles, CA.





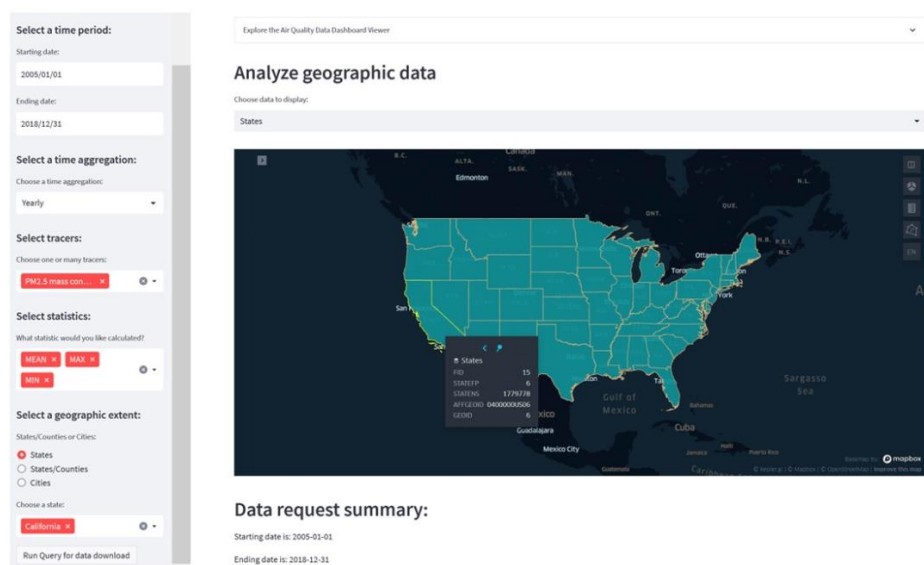


**Figure 13:** Streamlit Air Quality App to easily download and summarize data in a CSV format.






**9.    Appendices**
**A1: Forward and Adjoint operators for MOPITT CO assimilation**
MOPITT retrieved profile consists of 10 levels, including a surface level followed by 100 hPa
thick layers from 900 hPa to 100 hPa. The CMAQ vertical profile of CO cannot be compared with
MOPITT CO directly and needs to be convolved with the MOPITT a priori profile and averaging
kernel. Following (Barré et al., 2015; Gaubert et al., 2016), the CMAQ profile that can be
compared directly to MOPITT can be written as:
$CO_{ret}^{CMAQ} = 10^{(AK^{MOPITT} log_{10}(CO^{CMAQ}) + (I - AK^{MOPITT}) log_{10}(CO_{apr}^{MOPITT}))}$         (1)

$CO_{ret}^{CMAQ}$ is the CMAQ CO profile convolved with MOPITT a priori averaging kernel ($AK^{MOPITT}$)
and a priori profile ($CO_{apr}^{MOPITT}$) that can be compared directly to the MOPITT retrieved CO profile.
$CO^{CMAQ}$ is the 10-layer CMAQ profile mapped to the MOPITT pressure grid. A
$log_{10}$ transformation is necessary because the averaging kernel matrix for retrievals is obtained
with CO parameters in $log_{10}$(CO). Differentiation of equation (1) will yield the sensitivity of
$CO_{ret}^{CMAQ}$ with respect to $CO^{CMAQ}$, which represents the adjoint of the forward operator. For the
purpose of derivation, let $CO_{ret}^{CMAQ} = y$; $CO^{CMAQ} = x$; $AK^{MOPITT} = A$; and $(I -$
$AK^{MOPITT}) log_{10}(CO_{apr}^{MOPITT}) = C$ then equation (1) can be written as:
$y = 10^{(A log_{10}(x) + C)}$                 (2)
Applying the differentiation rule $\frac{d[a^u]}{dx} = ln(a) \cdot a^u \cdot \frac{du}{dx}$; we can differentiate equation (2) as:
$\frac{dy}{dx} = ln(10) \cdot 10^{(A log_{10}(x) + C)} \cdot \frac{d}{dx}(A log_{10}(x) + C)$         (3)
Since $A$ and $C$ do not depend on CMAQ simulations, they are constants and thus their
differentiation is zero. Since $\frac{d}{dx}(log_{10}(x)) = \frac{1}{x\,ln(10)}$, equation (3) simplifies to



$\frac{dy}{dx} = 10^{(A log_{10}(x) + C)} \cdot A \cdot \frac{1}{x} = A \cdot \frac{y}{x}$        (4)
Substituting the values of $y, x, A, and\ C$ in equation (4), the changes in CO vertical profile in the
MOPITT space can be related to changes in CO vertical profile in CMAQ as follows:
$dCO_{ret}^{CMAQ} = AK^{MOPITT} \cdot \frac{CO_{ret}^{CMAQ}}{CO^{CMAQ}} dCO^{CMAQ}$        (5)
By writing equation (5) in matrix form and then transposing the forward operator matrix, we can
write the adjoint of the forward operator as a recursive matrix equation:
$dCO^{CMAQ} = dCO^{CMAQ} + AK^{MOPITT} \cdot \frac{CO_{ret}^{CMAQ}}{CO^{CMAQ}} dCO_{ret}^{CMAQ}$        (6)


**A2: Additional Figures**

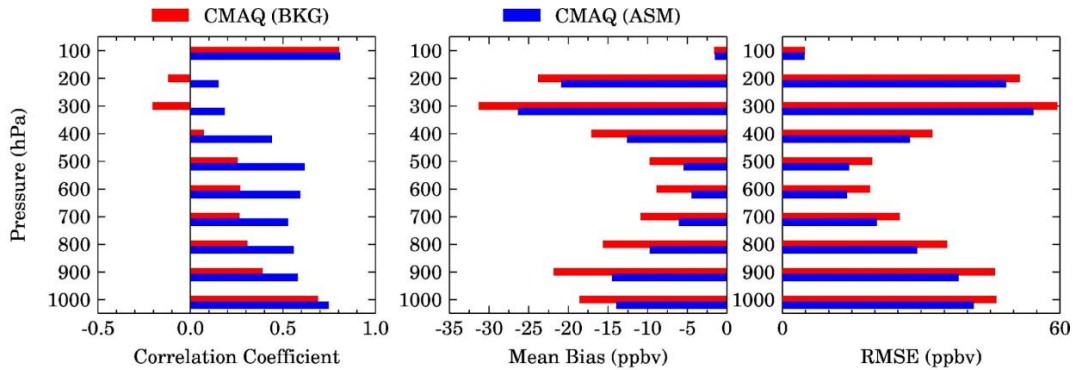


**Figure A2.1:** Correlation coefficient, Mean bias, and Root Mean Squared Error (RMSE)
between CMAQ and MOPITT CO profiles at ten MOPITT retrievals pressure levels for the
CMAQ experiments with (ASM) and without (BKG) assimilation of the MOPITT CO profiles
during July 2018. These statistics are based on 118552 data points at each level.







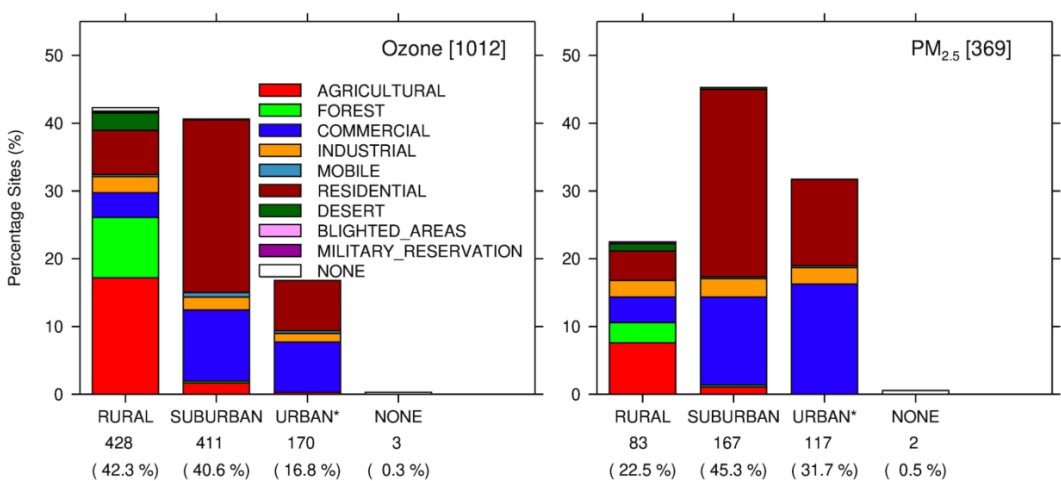

**Figure A2.2**: The stacked histogram shows the number of sites in each location setting (different

bars) and land use type (different colors) for MDA8 ozone (left) and 24-hr avg. PM$_{2.5}$ (right).

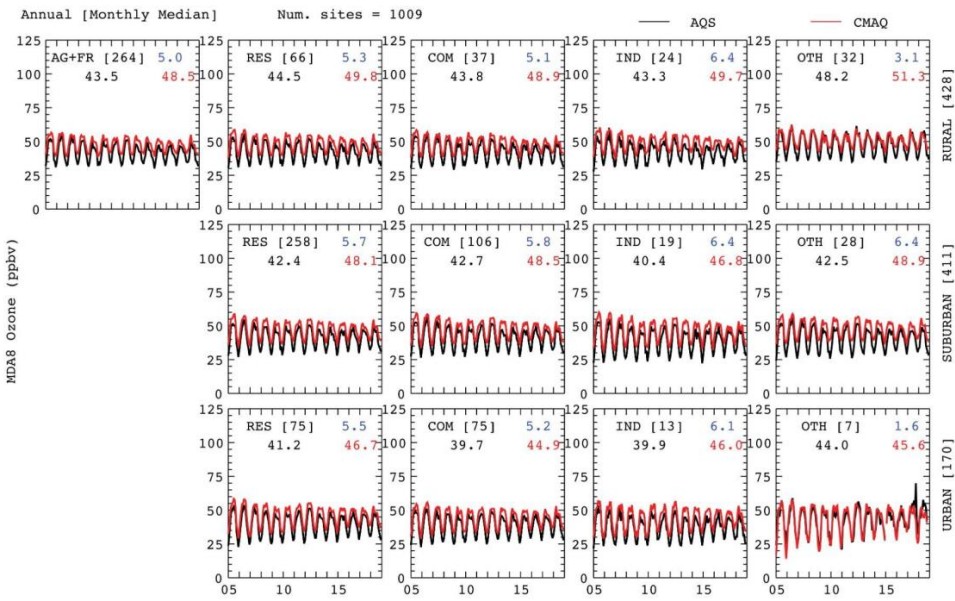

**Figure A2.3:** The Annual mean (derived from monthly median values) time series of MDA8

Ozone using AQS data (black) and CMAQ (red) over different location type (top to bottom) and

land-use type (left to right) during 2005-2018. The number of sites for each scenario are presented

in brackets. The blue color represents the mean bias.





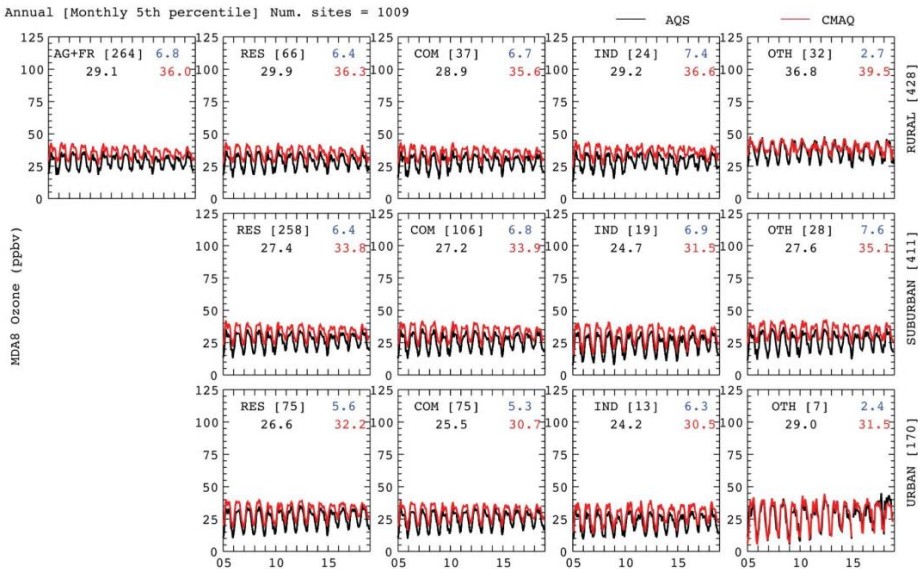


**Figure A2.4:** Same as Figure A2.3 but time series is derived from monthly 5th percentile values






Earth System Discussions
Open Access Science
Data

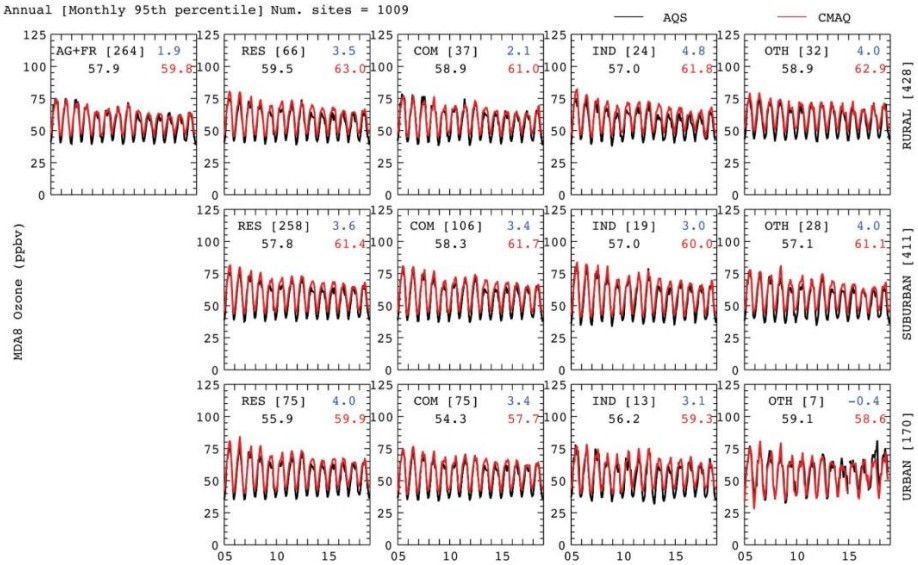


**Figure A2.5:** Same as Figure A2.3 but time series is derived from monthly 95[th] percentile values.



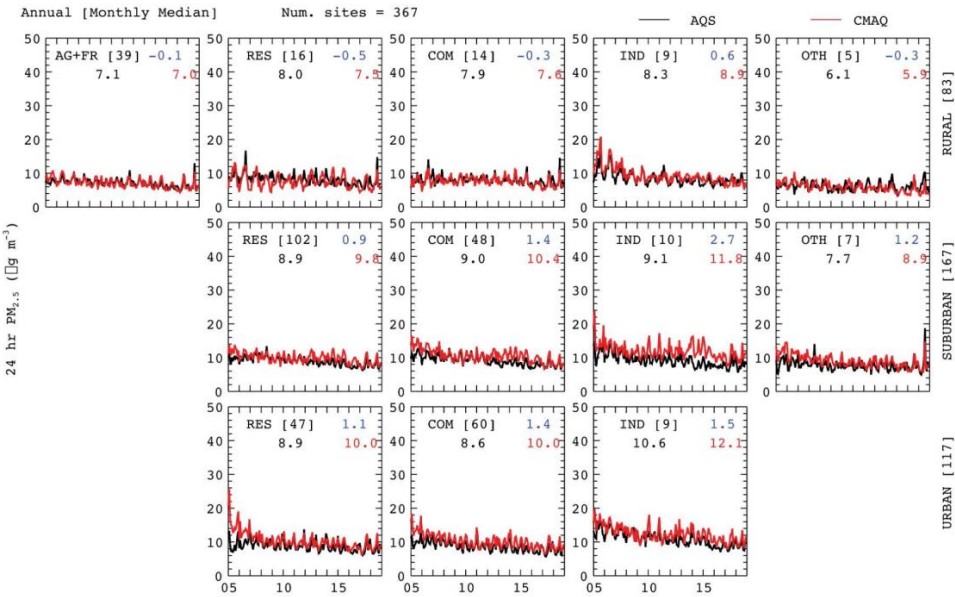

**Figure A2.6:** The Annual mean (derived from monthly values) time series of 24-hour avg. $PM_{2.5}$

using AQS data (black) and CMAQ (red) over different location types (top to bottom) and land-

use type (left to right) during 2005-18. The number of sites for each scenario are presented in

brackets. The blue color represents the mean bias.



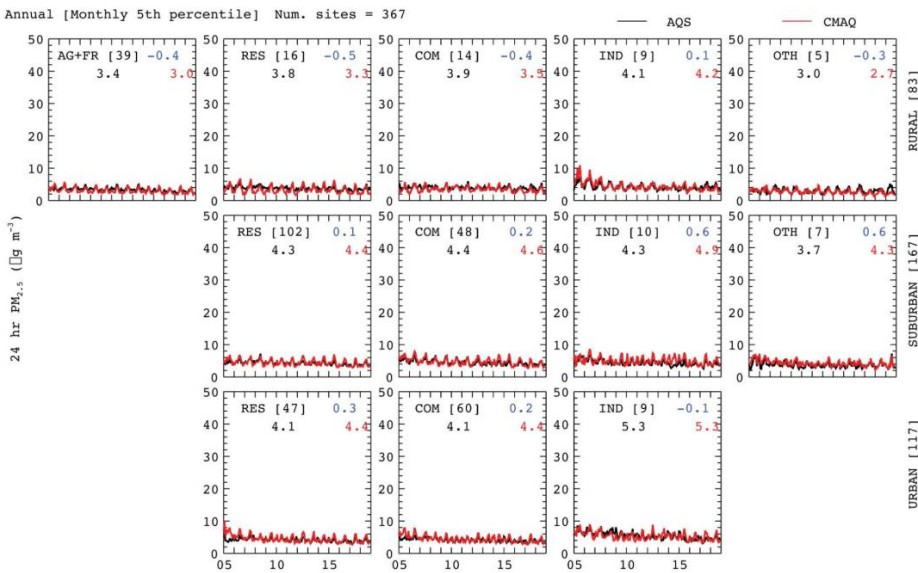


**Figure A2.7:** Same as Figure A2.6 but time series is derived from monthly 5th percentile values.



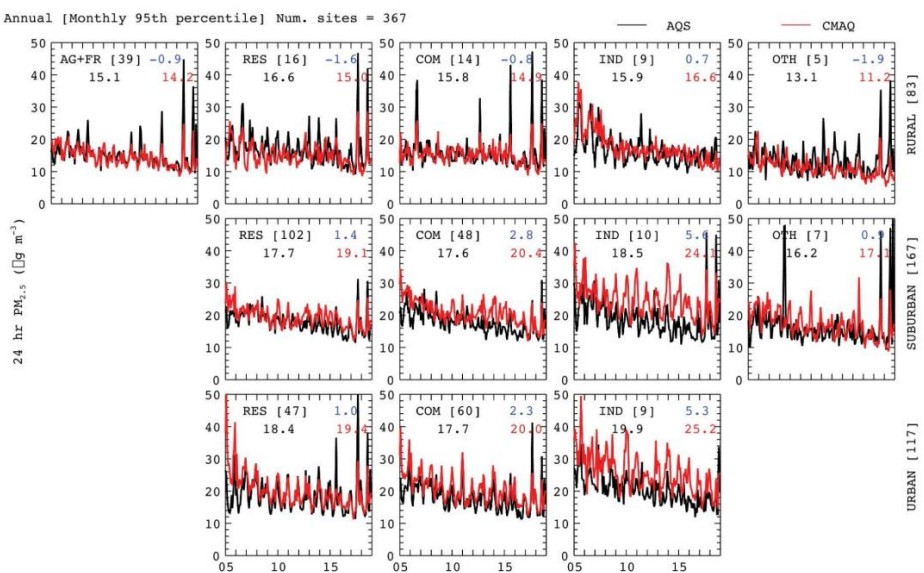


**Figure A2.8:** Same as Figure A2.6 but time series is derived from monthly 95th percentile values.

















**Table A3.1:** Key physics and chemical schemes used in the WRF-CMAQ configuration.

| Physics | Setup-1 (standard simulation used for assimilation) | Setup-2 (sensitivity simulation used to generate background error) |
|---|---|---|
| Long-wave radiation | RRTMG | RRTM Longwave |
| Short-wave radiation | RRTMG | Goddard Shortwave |
| Microphysics | Morrison double-moment | Thomson |
| Cumulus | Kain–Fritsch version 2 | Grell 3-D ensemble |
| Land surface model | Pleim–Xiu LSM | Unified Noah LSM |
| Surface Layer | Pleim–Xiu surface layer | MYNN |
| PBL | ACM2 | MYNN level 2.5 |
| Gas-phase chemistry | CB06 | CB06 |
| Aerosol chemistry | AERO7 | AERO7 |
| Anthropogenic and fire emissions | EPA NEI | EPA NEI perturbed by factors* derived from uncertainty analysis of multiple emission datasets |
| Biogenic emission | Online CMAQ BEIS | Offline MEGAN |






**Table A3.2:** Annual anthropogenic emissions for nine species over CONUS during 2005-2018.

| Emissions (Tg/yr) | HTAP v2 [2010] | EDGAR v4.3.2 [2010] | MACCity [2005-16] | CAMSv4.2 [2005-16] | NEI+ [2014] | Min-Max Ratio |
|---|---|---|---|---|---|---|
| CO | 56.20 | 56.77 | $46.02 \pm 6.39$ | $56.49 \pm 6.46$ | 45.69 | **1.24** |
| NH$_3$ | 4.42 | 5.14 | $4.44 \pm 0.14$ | $5.12 \pm 0.07$ | 3.25 | **1.58** |
| NO$_x$ | 11.07 | 10.93 | $10.40 \pm 1.00$ | $10.46 \pm 0.96$ | 12.03 | **1.16** |
| SO$_2$ | 13.10 | 12.52 | $10.87 \pm 2.44$ | $11.48 \pm 1.90$ | 4.46 | **2.94** |
| CH$_2$O | 0.12 | 0.20 | $0.17 \pm 0.02$ | $0.26 \pm 0.02$ | 0.16 | **2.17** |
| NMVOC | 15.61 | 14.57 | $6.58 \pm 0.82$ | $14.92 \pm 0.74$ | 12.28 | **2.37** |
| OC | 0.61 | 0.36 | $0.48 \pm 0.08$ | $0.36 \pm 0.01$ | 0.79** | **2.19** |
| BC | 0.34 | 0.20 | $0.28 \pm 0.06$ | $0.21 \pm 0.02$ | 0.26** | **1.70** |
| PM$_{2.5}$ | 2.02 | N/A | N/A | N/A | 3.67 | **1.82** |

[+] Except NEI, all other emissions are simply summed over {20-50 N} & {60-130 W} region
** CONUS PM$_{2.5}$ emissions are 5.15 Tg/yr which has 8% BC (or EC) and 28% OC
https://www.epa.gov/sites/production/files/2019-08/documents/210pm_rao_508_2.pdf




**Table A3.3:** Annual biomass burning emissions for nine species over the CONUS during 2005-

716 2018.

| Emissions (Tg/yr) | Top-Down emissions | | Bottom-up emissions | | | Min-Max Ratio |
|---|---|---|---|---|---|---|
| | QFED | GFASv1.3 | FINNv1.5 | GFEDv4.1 | NEI | |
| CO | 12.90 ± 2.59 | 8.99 ± 2.40 | 10.93 ± 2.21 | 5.41 ± 1.12 | 16.95 | 3.13 |
| NH₃ | 0.56 ± 0.11 | 0.12 ± 0.03 | 0.18 ± 0.04 | 0.07 ± 0.02 | 0.27 | 8.00 |
| NOₓ | 0.56 ± 0.11 | 0.20 ± 0.06 | 0.47 ± 0.10 | 0.18 ± 0.04 | 0.25 | 3.11 |
| SO₂ | 0.32 ± 0.07 | 0.07 ± 0.02 | 0.09 ± 0.02 | 0.04 ± 0.01 | 0.13 | 8.00 |
| CH₂O | 0.16 ± 0.03 | 0.15 ± 0.04 | 0.15 ± 0.03 | 0.10 ± 0.02 | 0.22 | 2.20 |
| tVOC | 0.53 ± 0.11 | 1.05 ± 0.28 | 1.86 ± 0.40 | 1.06 ± 0.22 | 3.92 | 7.40 |
| OC | 2.99 ± 0.63 | 0.60 ± 0.17 | 0.66 ± 0.13 | 0.34 ± 0.09 | 0.45 | 8.79 |
| BC | 0.24± 0.05 | 0.05 ± 0.02 | 0.06 ± 0.01 | 0.03 ± 0.01 | 0.15 | 8.00 |
| PM2.5 | 4.37 ± 0.92 | 0.90 ± 0.24 | N/A | 0.61 ± 0.14 | 1.48 | 7.16 |




**10. Author contribution**
RK, PB, CH, GGP, SA, HW, and OG conceptualized the study. All the authors contributed to the
design of the study. RK, CH, and PB performed all the model simulations including the data
assimilation system developments and experiments. PB, CH, RK, and SA contributed to the model
evaluation and trend analysis. RK, FL, JB, OG, KS, MC, and SS contributed to the design of the
air quality dashboard and Streamlit application. RK prepared the first draft of the paper. All authors
contributed to the editing of the manuscript.

**11. Competing interests**
The authors do not have any competing interests. The funding agency had no role in the design of
the study, in the collection, analyses, interpretation of data, in the writing of the manuscript,
or in the decision to publish the results.

**12. Financial support**
This work is supported by the NASA Atmospheric Composition Modeling and Analysis
(ACMAP) Grant # 80NSSC19K0982.

**13. Acknowledgement**
We would like to acknowledge high-performance computing support from Cheyenne (doi:
10.5065/D6RX99HX (accessed on 08 June 2023)), provided by NCAR's Computational and
Information Systems Laboratory, sponsored by the National Science Foundation. The National
Center for Atmospheric Research is sponsored by the National Science Foundation under
Cooperative Agreement 1852977.

—





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
