# Peer review of "A long-term high-resolution air quality reanalysis with public facing air quality dashboard 1 2 over the Contiguous United States (CONUS) 3 Rajesh Kumar1, Piyush Bhardwaj1,\*, Cenlin He1, Jennifer Boehnert1, Forrest Lacey1, Stefano 4 Alessandrin"

_Earth System Science Data, 2024_

## Author Comment (AC1)

**Responses to Reviewer #1**

The paper assimilated a long-term data which is critical for air quality managers and researchers to understand the long term air quality trends and changes at a fine scale county level. Also made that long-term data available to the stakeholders through a dashboard which makes it very easy to visualize the data.

We thank the reviewer for a thorough review of our manuscript and for providing constructive comments. Below, we provide a point-by-point response to your concerns. Reviewer's comments appear in regular black font and our responses appear in regular blue font.

Specific Comments:

R1.1) Line 107: Is the methodology any different than any of the previous studies? If so the author should provide that information and highlight the differences.

A1.1) Our regional reanalysis is based on three-dimensional variational (3DVAR) approach, which is different compared to the four-dimensional variational (4D-Var) approach (Inness et al., 2019) and Ensemble Kalman Filter approaches (Gaubert et al., 2017; Kong et al., 2021; Miyazaki et al., 2020) used in recent long-term global and regional air quality reanalysis. Among these, 3DVAR is computationally the most efficient approach because it uses only a single model simulation, but its accuracy can be limited by the assumption of a constant background error covariance matrix that both 4DVAR and EnKF address. This information is added to the revised manuscript.

References:

Gaubert, B., Worden, H. M., Arellano, A. F. J., Emmons, L. K., Tilmes, S., Barré, J., Alonso, S. M., Vitt, F., Anderson, J. L., Alkemade, F., Houweling, S., and Edwards, D. P.: Chemical Feedback From Decreasing Carbon Monoxide Emissions, Geophys. Res. Lett., 44, 9985–9995, https://doi.org/10.1002/2017GL074987, 2017.

Inness, A., Ades, M., Agustí-Panareda, A., Barré, J., Benedictow, A., Blechschmidt, A.-M., Dominguez, J. J., Engelen, R., Eskes, H., Flemming, J., Huijnen, V., Jones, L., Kipling, Z., Massart, S., Parrington, M., Peuch, V.-H., Razinger, M., Remy, S., Schulz, M., and Suttie, M.: The CAMS reanalysis of atmospheric composition, Atmospheric Chem. Phys., 19, 3515–3556, https://doi.org/10.5194/acp-19-3515-2019, 2019.

Kong, L., Tang, X., Zhu, J., Wang, Z., Li, J., Wu, H., Wu, Q., Chen, H., Zhu, L., Wang, W., Liu, B., Wang, Q., Chen, D., Pan, Y., Song, T., Li, F., Zheng, H., Jia, G., Lu, M., Wu, L., and Carmichael, G. R.: A 6-year-long (2013–2018) high-resolution air quality reanalysis dataset in China based on the assimilation of surface observations from CNEMC, Earth Syst. Sci. Data, 13, 529–570, https://doi.org/10.5194/essd-13-529-2021, 2021.

Miyazaki, K., Bowman, K., Sekiya, T., Eskes, H., Boersma, F., Worden, H., Livesey, N., Payne, V. H., Sudo, K., Kanaya, Y., Takigawa, M., and Ogochi, K.: Updated tropospheric chemistry reanalysis and emission estimates, TCR-2, for 2005–2018, Earth Syst. Sci. Data, 12, 2223–2259, https://doi.org/10.5194/essd-12-2223-2020, 2020.

R1.2) Line 188: What is the rationale to perform 9 simulations every day rather than the every hour of the day?

A1.2) This is because of the three-hour difference between Terra and Aqua overpass times. Since we are assimilating only satellite observations, there are no unique observations available for assimilation every hour. We have added this information to the revised manuscript.

R1.3) Line 189: The first simulation seems to cover more time period that the subsequent simulations. Will that compromise any of the model predictions?
A1.3) No, this does not compromise any model predictions because there are no satellite retrievals available over CONUS before 15 Z.

R1.4) Line 212: Did the author perform any trace gas species performance for the with and without assimilation of the MOPITT to show the indirect effect of CO on the trace gas species?
A1.4) Yes, we analyzed the impact of MOPITT CO assimilation on surface ozone because CO is a precursor of ozone and also affects the oxidative capacity of the atmosphere. We found instantaneous changes in the range of -1.3 to 3.2 ppbv but monthly average changes are within the range of ±0.3 ppbv in surface ozone. This information has also been added to the revised manuscript.

R1.5) Line 275: Can the author specify the 10 regions in the text or in the supplementary document? The 10 regions information will be helpful for the reader to understand the regional changes in the data.
A1.5) We have included a map showing the 10 EPA regions in appendix A2 as Figure A2.2 in the revised manuscript. The map is reproduced below in Figure R1.1 for your reference. Note that our evaluation does not include Puerto Rico in Region 2, Hawaiian Islands in Region 9, and Alaska in Region 10.

[Figure]

**Figure R1.1:** Map showing the EPA regions over which model evaluation has been performed. The map is reproduced from https://www.epa.gov/aboutepa/visiting-regional-office.

R1.6) Line 279: Figure 2-5: The time series for all the parameters appears too clustered and it is not clear to see the red and black lines. Is it possible to reduce the temporal resolution to something like daily while plotting?

A1.6) The plots looked very clustered even with the daily averages. So, we have changed these plots to monthly average values with standard deviation and tried to increase the transparency of these plots (see Figure R1.2 below as an example for 2 m temperature). Following reviewer #2 suggestions, we have also added plots showing diurnal variations in T2, RH, and 10 m wind speed as Figure A2.3 in the revised manuscript (see Figure R1.3 below).

[Figure]

**Figure R1.2:** Time series of monthly averaged 2 m temperature over 10 EPA regions (R1-R10) from WRF-CMAQ setup (red-triangle) and METAR observations (black-circle) during 2005-2018. Orange and Grey lines represent the standard deviation for WRF-CMAQ and METAR, respectively. The correlation coefficient (r), mean bias (MB), and the root mean square error (RMSE) for each region is also shown.

[Figure]

**Figure R1.3:** Seasonal mean diurnal variations in 2 m Temperature (Top panel), relative humidity (middle panel) and 10 m wind speed (bottom panel) from METAR observations and WRF model.

R1.7) Line 323: Figure 7: Again the hourly time series looks very overcrowded, can this be changed to daily MDA8?

A1.7) We have added daily MDA8 and Daily (24-hr) mean PM2.5 as figs. 7, 8. The plots are also shown below for reference in Figures R1.4 and R1.5, respectively.

[Figure]

**Figure R1.4**: Time series of daily Maximum Daily 8-hour average (MDA8) surface ozone over 10 EPA regions (R-01 to R-10) from WRF-CMAQ setup (red) and EPA AQS observations (black) during 2005-2018. The correlation coefficient (r), mean bias (MB), and the root mean square error (RMSE) for each region are also shown.

[Figure]

**Figure R1.5**: Same as Fig. R1.4 but for daily 24-hour average PM$_{2.5}$

R1.8) Line 325-327: The author made a point about the nighttime ozone but there is no figure or data to support the statement. Can the author include figure or a table in the appendix to show this finding?
A1.8) We have added seasonal averaged diurnal profiles shown in Figure R1.6 for your reference. This figure has been added in Appendix A2 Figure A2.4 in the revised manuscript.

[Figure]

**Figure R1.6:** Average diurnal profile of ozone (top panel) and PM$_{2.5}$ (bottom panel) over all AQS sites in CONUS.

R1.9) Line 373: Explain what is trend analysis here, so that the reader can understand the context clearly.
A1.9) Thanks for this suggestion. We have added the following information now: "To help air quality managers and the public determine the confidence they can put in using this reanalysis for analyzing changes in air quality in their regions, we have evaluated the trends in our CMAQ simulated MDA8 ozone and 24-hr average PM$_{2.5}$ against the AQS observations."

R1.10) Line 379: Why did the author pick 2-sigma level for this analysis?
A1.10) In a normal distribution, ~95% of the data points lie within 2 standard deviations (±2$\sigma$) of the mean. If the trend falls outside this range, it is considered unlikely to have occurred by chance (i.e., at a statistical significance in the probability of less than 5%). Therefore, the 2-sigma rule is a standard way of testing whether a trend is statistically significant. This information has been added to the revised manuscript.

R1.11) Line 391: Include the number of sites in the parenthesis.
A1.11) We apologize for the typo in the percentage number given in this line. Lines 389-392 have been changed to include information about the number of sites. Here is the revised text: "About 55% (278 of 501) of the sites showed positive trends in both AQS and CMAQ data during winter but only ~3% (29 of 1012) of the sites showed positive trends in summer."

R1.12) Line 509: Reword this line "millions of people in counties with no monitors"

A1.12) The line has been revised to "Air pollution is an important health hazard affecting human health and the economy in the CONUS, yet millions of people live in counties without air quality monitors."